# High-Level Smart Decision Making of a Robot Based on Ontology in a Search and Rescue Scenario

**Xiaolei Sun, Yu Zhang * and Jing Chen**

College of Intelligence Science and Technology, National University of Defense Technology,
Changsha 410073, China; sunxiaoleijc@163.com (X.S.); chenjingnudt@163.com (J.C.)
* Correspondence: redarmy_zy@163.com

**Abstract:** The search and rescue (SAR) scenario is complex and uncertain where a robot needs to understand the scenario to make smart decisions. Aiming at the knowledge representation (KR) in the field of SAR, this paper builds an ontology model that enables a robot to understand how to make smart decisions. The ontology is divided into three parts, namely entity ontology, environment ontology, and task ontology. Web Ontology Language (OWL) is adopted to represent these three types of ontology. Through ontology and Semantic Web Rule Language (SWRL) rules, the robot infers the tasks to be performed according to the environment state and at the same time obtains the semantic information of the victims. Then, the paper proposes an ontology-based algorithm for task planning to get a sequence of atomic actions so as to complete the high-level inferred task. In addition, an indoor experiment was designed and built for the SAR scenario using a real robot platform—TurtleBot3. The correctness and usability of the ontology and the proposed methods are verified by experiments.

**Keywords:** ontology; search and rescue; smart decision-making; task planning; knowledge representation

## 1. Introduction

Today, many theories and methods have emerged in the field of artificial intelligence (AI), which have also been deeply applied in many domains. As a typical method of AI, deep learning (DL) has made a great breakthrough [1]. It is also widely used in robotics [2,3]. However, DL is difficult to explain, which limits its application in some fields requiring knowledge reasoning. Since the 1970s, AI researchers have gradually realized that symbolic knowledge methods play a key role in more powerful AI systems. They think that knowledge and knowledge reasoning are the core of AI. Since then, ontology has been strongly developed as a form of knowledge base. It can represent and understand the complex real world. Similar to the human thinking mode, robots can use knowledge and reasoning to realize smart decision-making. Now, it has been widely used in AI [4,5], semantic web [6,7], informatics [8], and other fields.

There are many kinds of complex emergency tasks in disaster search and rescue scenario. It is a huge challenge for a robot to perform these kinds of complex tasks. After a disaster, the robot must explore the whole unknown environment, locate the victims, and send back the information of the victims in a short time [9–11]. An important goal of the article is to make robots capable of performing search and rescue (SAR) tasks autonomously. However, at the present stage, all rescue robots are remotely controlled by operators in the actual rescue tasks [12]. In this way, it is easy to lose direction, bump into obstacles by mistake, or fall into negative obstacles [13]. In order to reduce the operation errors of rescue workers, it is necessary to improve the autonomy and intelligence of robots. High autonomy can greatly improve efficiency, and high intelligence can improve the performance of robots.

As shown in Figure 1, the control of robots can be divided into three levels. Low-level control focuses on the control of motors and sensors. Middle-level control focuses on environment perception, Simultaneous Localization and Mapping (SLAM), and autonomous navigation. High-level control mainly involves smart decision-making of robots. In previous studies, most researchers focused on SLAM and navigation, corresponding to the middle-level control in Figure 1. In this level, search and rescue are not targeted at the beginning, and the robot generally needs to explore the environment globally, with low efficiency and lack of prior knowledge of SAR. Our research focuses on high-level control and smart decision-making of robots based on ontology in an SAR scenario.

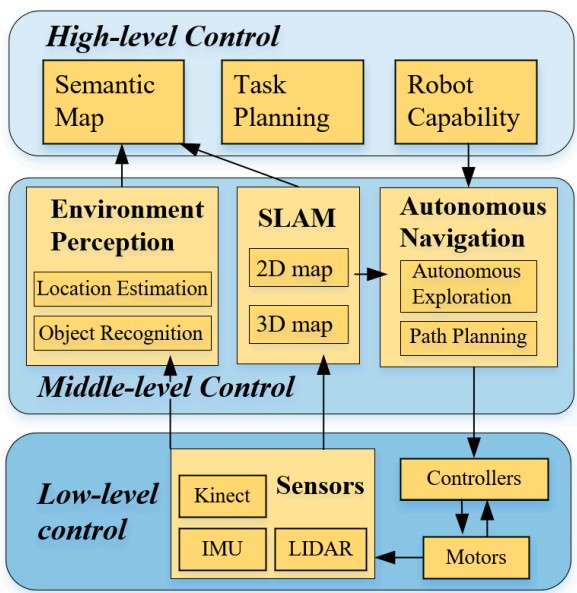

**Figure 1.** The three levels of robotic control defined in article.

There are some typical study works and applications on robotic ontology. KnowRob [14–16] is an integrated knowledge management system in autonomous robots, which aims to build a knowledge base for indoor service robots. It consists of Web Ontology Language (OWL) language ontology and an extensible reasoning engine. However, it is suitable in the indoor service environment and difficult to extend to the application of SAR. In addition, OpenRobots Common-Sense ontology (ORO) [17,18] integrates data from a variety of sources, such as sensors and human interaction. After being built in OWL, ORO is stored in the OpenJena ontology management library. However, ORO focuses on human–computer interaction and lacks support for SAR. Smart and Networking Underwater Robots in Cooperation Meshes (SWARMs) Ontology [19–21] is designed to describe and understand the complex environments of underwater unmanned robots and promote multi-robot cooperation. However, it is only built for underwater unmanned robots. The paper by Sadik and Urban [22] applied the ontology methods to industrial production. The action schema (corresponding to the task ontology presented in our article) is built in the form of action recipes. When decomposing the high-level task, the program directly queries the action recipes, which lack extendibility and intelligence. Perception and Manipulation Knowledge (PMK) [23] puts forward a knowledge processing framework for navigation tasks in indoor scenario following IEEE-1872 standards [24]. In references [22,25], they use other logical languages, such as BC, to represent knowledge in the field of robotics [26,27]. However, this approach is not conducive to the sharing of knowledge and the updating of new knowledge. However, there has not been an ontology built for SAR. Our ontology is designed especially for SAR and is applied to the high-level decision-making of robots. Task planning is easy with the knowledge representation in SAR.

The challenge of our research work is how to efficiently and reasonably represent and update the complex task knowledge for smart decision-making in SAR. On the basis of the ontology we have

built, another challenge is how to infer the tasks the robot should to perform in its current state. Then, the robot should know how to decompose the tasks into a sequence of atomic actions.

We consider spatio-temporal, continuous, and discrete information to represent the knowledge in SAR with the form of ontology. The Semantic Web Rule Language (SWRL) was used to compensate for OWL's inability to express complex rules. The robot can reason through the ontology with the SWRL rules to obtain the tasks the robot should perform in the current circumstances. Besides, in the process of performing tasks, the robot can obtain the position information of victims by Bayesian Reasoning and get other semantic information by scanning the QR (Quick Response) code. All the information will then be updated in the ontology. In the ontology, the representation of high-level tasks and atomic actions are independent. This article proposed an algorithm for task planning in SAR by matching the execution preconditions of atomic actions and their effects on the environment from the initial state to the goal. The task planning module can also be executed to adapt the different input tasks. Finally, the indoor experiment environment is built, and the usability of the built ontology is verified experimentally. In the real robot experiment, we used Robot Operating System (ROS) [28] as the software system and TurtleBot3 as the hardware platform. The experimental results show that the robot can perform the high-level smart decision-making well based on the ontology. The ontology we have built provides a domain knowledge base for the robot to better understand a complex environment and plan tasks in an SAR scenario.

The rest of the paper is arranged as follows. Section 2 describes the ontology we have built in detail. Section 3 presents the methods we adopted about the knowledge reasoning and knowledge updating. Section 4 introduces the methods of carrying out the experiment about smart decision-making by the real robot. Section 5 displays the results obtained from the experiments. Section 5 also analyzes and discusses the results. Finally, Section 6 summarizes all the work and highlights future research directions.

## 2. Ontology in SAR Scenario

In this section, we will introduce the ontology in the SAR scenario we have built in detail. First, we give the definition of the elements involved in the ontology. Then, this section describes the ontology we built from three parts.

### 2.1. The Definition

In high-level decision-making, the tasks and actions are the core part to achieve some goals. In addition, all the parts can be connected through tasks and actions. Therefore, we define the following two definitions about tasks and actions.

**Definition 1.** *A high-level task model is defined as a 5-tuple.*

$$\mathcal{T} = \left( T_{name}, T_{attr}, T_{entity}, T_{tasker}, T_{methods} \right) \tag{1}$$

Consisting of a set of task names, $T_{name}$, a set of task attributes, $T_{attr}$, e.g., the start time and initial state, a set of entities carrying out the tasks, $T_{entity}$, a set of taskers giving the high-level tasks, $T_{tasker}$, and a set of methods to decompose the high-level task, $T_{methods}$.

**Definition 2.** *An atomic action model is defined as a 5-tuple.*

$$\mathcal{A} = \left( A_{name}, A_{attr}, A_{entity}, A_{pre}, A_{effect} \right) \tag{2}$$

Consisting of a set of atomic actions names, $A_{name}$, a set of atomic actions attributes, $A_{attr}$, a set of entities carrying out the atomic actions, $A_{entity}$, a set of execution preconditions of atomic actions, $A_{pre}$, and a set of actions effects $A_{effect}$.

### 2.2. The Ontology in SAR

The term "ontology" derives from metaphysical concepts in the field of philosophy, and there is no unified definition [29]. In the research field of computer science, ontology represents a set of concepts, relations, attributes, and instances in the world. It can also reason about new or hidden knowledge based on known factual knowledge. Ontologies are widely used in many domains, such as information science [30,31], medical science [32,33], and education [8,34].

The ontology in our research work is divided into a conceptual layer and an instance layer. The conceptual layer can be divided into an abstract layer and a specific layer. As shown in Figure 2, the conceptual layer describes the common conceptual knowledge related to the disaster rescue domain. The abstract layer includes the high-level and more abstract concepts, such as the SLAM class, object class, task class, and so on. The specific layer involves the concepts that are defined as more specific kinds, such as environment objects (e.g., furniture). The instance layer is the instantiation of conceptual layer and the relationships that exist among the instances. The relationships connect the instances with each other and make intelligent behavior possible.

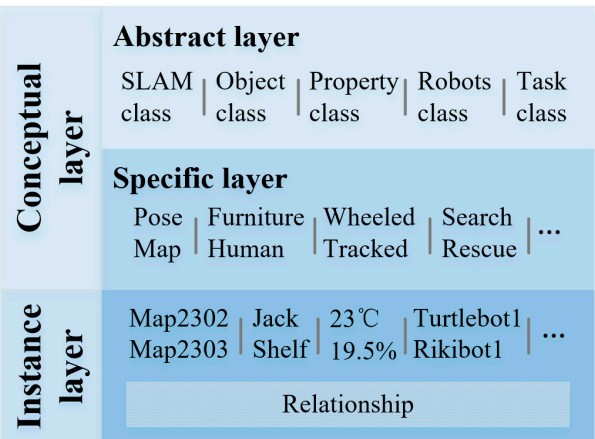

**Figure 2.** The classification of different layers in ontology.

The ontology contains three parts: entity ontology, environment ontology, and task ontology. Task ontology describes the task knowledge related to smart decision-making of robots, such as the task decomposition and task allocation via hierarchical structure. Entity ontology is designed to portray the knowledge or concepts of entity corresponding to the robot. Finally, environment ontology gives a description of the knowledge related to the SAR scenario such as the environment map, environment objects information, and so on.

As displayed in Figure 3, the paper extends the task ontology from four typical tasks, which are charge task, search task, rescue task, and recognize task. They are involved in the subsequent experiment to verify the designed algorithms. The task ontology is hierarchical seen in Figure 3. The task has subclass of the subtasks and atomic actions. Going forward, the preconditions of each task and atomic actions are defined with the data properties in Protégé. Besides, the atomic actions are defined to have effects (delete and add) on the environment state. The task description structure is similar to Hierarchical Task Network (HTN). Thus, we can decompose the tasks the robot reasoned through ontology for task planning in SAR.

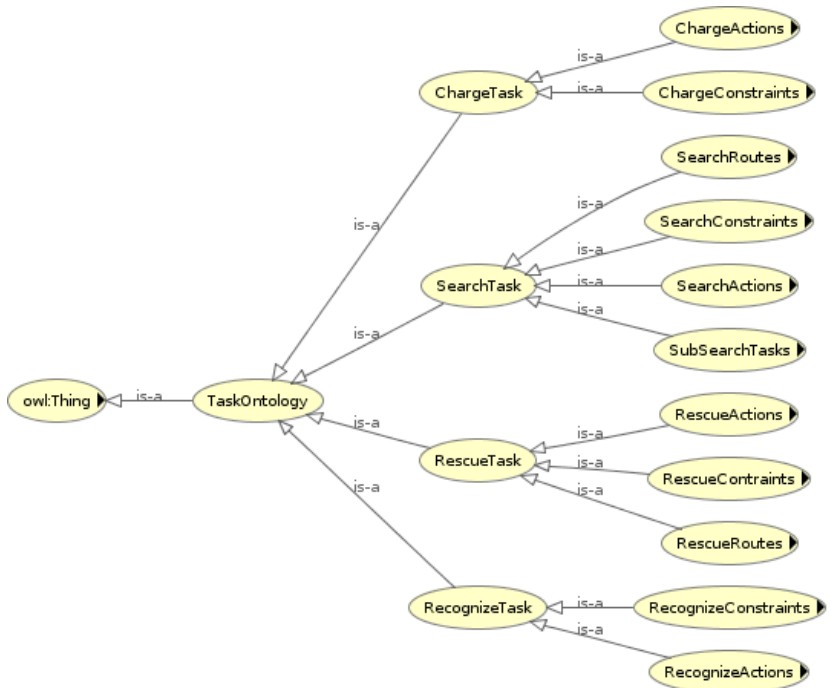

**Figure 3.** The task ontology is shown in Protégé.

When the robot performs a task for the first time, the initial representation of the task in OWL can be described as the follows. The task *RescueJack* contains three atomic actions: *PickUpVictim*, *GetOutVictim*, and *PutDownVictim*. The time sequence is unknown.

**Class**: RescueJack
   **SubClassOf:**
   SubRescueTasks
(subAction some FreeVictim)
and (subAction some PickUpVictim)
and (subAction some GetOutVictim)
and (subAction some PutDownVictim)

Then, the robot will query and match the task ontology layer by layer according to the current initial state and reason about the final sequence of atomic actions. The sequence of atomic actions contains the time sequence of action execution and is a complete representation of high-level input tasks. A complete representation of a specific task is shown as follows, which is composed of the parent classes, the subactions, and the execution ordering constraints among subactions. When a specific task requires planning, the users can query and search to obtain the atomic actions sequence of the specific task.

**Class**: RescueJack
   **SubClassOf:**
   SubRescueTasks
(subAction some FreeVictim)
and (subAction some PickUpVictim)
and (subAction some GetOutVictim)
and (subAction some PutDownVictim)
and (orderingConstraints value RescueActions12)
and (orderingConstraints value RescueActions13)

and (orderingConstraints value RescueActions14)
and (orderingConstraints value RescueActions23)
and (orderingConstraints value RescueActions24)
and (orderingConstraints value RescueActions34)

Relying on the individual class in Protégé, the one of constraints among subactions is defined as follows. It defines the time sequence of action *PickUpVictim* and action *FreeVictim*. Assuming that a specific task has *n* subactions, it is easy to know that the total number to define the specific task is $C_n^2$, provided that we completely define the time sequences of all subactions.

**Individuals**: RescueActions12
 **Types:**
 PartialOrdering-Strict
 **Annotations:**
 occursAfterInOrdering PickUpVictim
 occursBeforeInOrdering FreeVictim

As shown in Figure 4, the entity ontology contains three parts—robots, hardware, and software—which demonstrate the capability and characteristic of robots. Robots include various types of robot, like ground robots, underwater robots, and air robots. The concepts of robots can be instantiation as individuals in Protégé. Hardware is composed of the components and devices the various types of robots may have. It can be further divided into perception devices, navigation devices, and base devices. Navigation devices refer to the hardware devices that robots need to navigate and locate, such as IMU (Inertial Measurement Unit). Perception devices refer to the hardware devices that robots need to perceive and understand the environment, such as LIDAR (Light Detection and Ranging) and a camera. Base devices refer to the hardware devices related to the low-level control of the robot, such as the motor, battery, and so on. Software consists of the functional ROS node, which can publish its specific topic and subscribe to the other topics. It can also achieve the communication among control nodes. A variety of relationships can be defined by developers to describe the relationship among the entity ontology and with other ontologies.

As shown in Figure 5, the environment ontology is mainly used to query and reason how to make the high-level decision for the intelligence of robots. It is important for the achievement of intelligent robots. The environment ontology describes all the things existing in SAR. It includes the environment map, which is already known for the robots to navigate. It also includes the objects that may be unknown for the robots and need to be recognized to update the environment ontology. The updated knowledge can be shared with other robots to reduce the repetitive work and improve the efficiency of task execution.

Apart from the contents above, corresponding relationships also exist among the three parts which connect this knowledge with one another. These relationships can be defined according to the developers' own needs. Taking the disaster rescue task as an example (Figure 6), the rescue task is achieved by the mobile wheeled robot TurtleBot3 and in the environmental map of Room2. The latter is built by the Lidar of TurtleBot1. Accordingly, these three ontology modules can be linked and added to constraints so that they make up the whole ontology jointly.

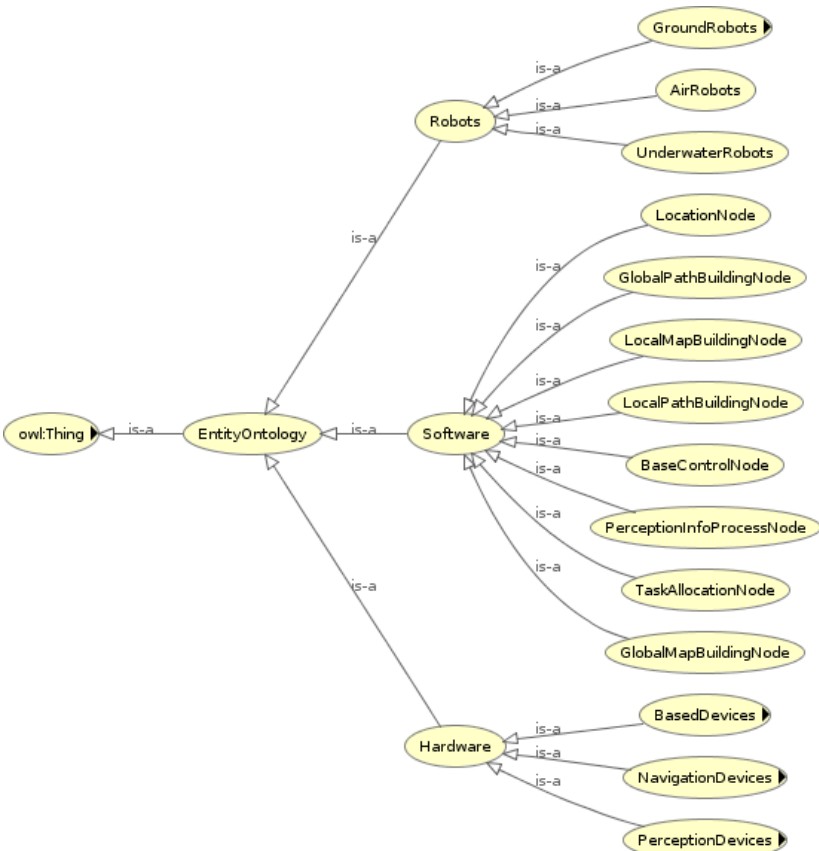

**Figure 4.** The entity ontology is shown in Protégé.

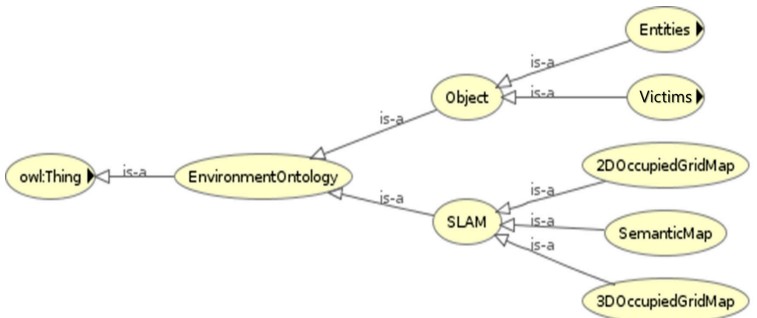

**Figure 5.** The environment ontology shown in Protégé.

It takes about 2.34 s to generate 55,000 individuals by the evaluation test of our ontology. The maximum generation rate is about 23,500 individuals per second. As a comparison, KnowRob [15] has a maximum generation rate of 22,000 individuals per second, and ORO [17] has a maximum generation rate of 7245 individuals per second.

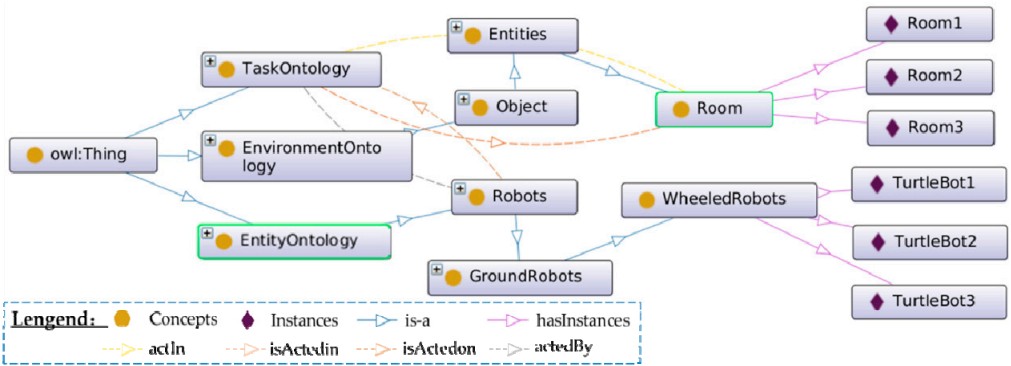

**Figure 6.** The relationships among three parts in Protégé.

## 3. Methods

This section mainly introduces how the system reasons and updates the ontology knowledge in the process of search and rescue based on the ontology we have built. Our work focuses on the high-level control and smart decision-making of robots. Therefore, in the SAR scenario, we assume that the robot first needs to reason about the task that is suitable for the current state according to the factual knowledge and SWRL rules. Next, the robot needs to decompose the task into a sequence of atomic actions and then perform the task. In the process of task execution, if the robot finds the target, it needs to infer the location of the target and some semantic information and update the ontology knowledge in a timely fashion. The following section describes the methods involved in this process.

### 3.1. Reasoning Based on SWRL Rules

OWL is a description language of ontology that can represent the complex systems and environments. The robots can obtain the complete factual knowledge based on the ontology built in Section 2. On the premise of correctly perceiving the SAR scenario, the robot needs to further realize the high-level task decision-making and task execution functions. Therefore, it is necessary to reason based on the knowledge of known facts to obtain new knowledge and infer the tasks needed to be completed to adapt to the current state. OWL has the disadvantage of lacking in describing general rules, especially the "if-else-then" statements. Because of that, the users cannot check the hidden knowledge with ontology in OWL. In this section, SWRL language is used to build rules base as the basis for ontology inference of robots. It is a promising approach to combine the SWRL and OWL to help OWL describe the general rules. The robot can obtain the task need to be completed to adapt the current state by the ontology inference engine JESS. Based on the obtained task and the task ontology, the task decomposition method will be described in detail in Section 3.2, which will generate the time sequences of atomic actions.

### 3.1.1. Structure and Syntax of SWRL Rules

SWRL is a type of language that describe rules in a semantic way. It is composed of OWLDL and OWLLite, which are the sub-languages of OWL. Besides, it follows the high-level abstract grammar of Horn rules and is one of the standards of W3C [35]. SWRL and OWL are highly integrated with each other, and rules in SWRL can be combined with knowledge in OWL. SWRL structure is mainly divided into four parts, namely *Imp*, *Atom*, *Variable*, and *Built-in*. The rules of SWRL consists of the part *Imp*, in which the rule is represented as the form *precondition* (body) → *conclusion* (head). *Imp* contains the head and body, in which head represents the conclusion of the inference rules, and body represents the precondition that the inference can be reached. The head and body are made up of the basic elements provided by *Atom* or *Variable*. The following is a specific rule to analyze SWRL, and the rule is

*Robot (?r) ^ hasBatteryQuantity (?r, ?batteryquantity) ^ lowerThan (?batteryquantity, 10%) -> needCharge(?r)*

Rules are built with OWL classes, properties, instances, and data values. The above rule declares the class *Robot* and *needCharge* and captures the property *hasBatteryQuantity* for inference. Using the function of the SWRL *Built_in* component, the battery quantity of robot is compared with the minimum battery quantity. The rule asserts that, if a robot's battery is less than 10 percent, it indicates that the robot needs to be charged.

3.1.2. The Constructions of SWRL Rules

The steps to build SWRL rules base in SAR are as follows:

- Extract the key rule knowledge of robot automatic search and rescue from relevant books, literature and manuals, and form the rule knowledge in the form of natural language;
- Declare this rule knowledge in the formal description language and specify the Precondition of the SWRL rule;
- Determine the types and instances of the concepts involved in the rule knowledge.

We take the initial state of SAR tasks as an example and design the following rule knowledge:

Rule_1: If the initial position of SAR robot is the center of SAR map, the robot will perform the center search route.
Rule_2: If the initial position of SAR robot is the Corridor of SAR map, the robot will perform the cross search route.
Rule_3: If the initial position of SAR robot is the Room of SAR map, the robot will perform the square search route.

For the above rule knowledge, we analyze the Atom of SWRL rule contained in the ontology as shown in Table 1.

**Table 1.** The Atom of Semantic Web Rule Language (SWRL) rules contained in the ontology.

| Atom | Description |
| --- | --- |
| hasAbility(?A, ?WR) | WR has the A ability |
| hasAction(?ST, ?WR) | WR performs the ST task |
| hasPosition(?WR, ?ER) | WR locates in ER |
| WheeledRobots(?WR) | WR is a wheeled robot |
| ChooseSearchRoute(?SR, ?ST) | the ST task choose the SR search route |
| Victims(?V) | V is a victim |
| Center(?ER) | ER is the center of SAR map |
| Corridor(?ER) | ER is the corridor of SAR map |
| Room(?ER) | ER is the room of SAR map |
| SquareSearch(?SR) | SR is the square search route |
| CrossSearch(?SR) | SR is the cross search route |
| CenterSearch(?SR) | SR is the center search route |
| SearchTask(?ST) | ST is the search task |

The established SWRL rules are as follows:

Rule_1: WheeledRobots(?WR) ^ hasAbility (?A, ?WR) ^ Center(?ER) ^ hasPosition(?WR, ?ER) ^ SearchTask(?ST) ^ CenterSearch(?SR) -> ChooseSearchRoute(?SR, ?ST)
Rule_2: WheeledRobots(?WR) ^ hasAbility (?A, ?WR) ^ Corridor(?ER) ^ hasPosition(?WR, ?ER) ^ SearchTask(?ST) ^ CrossSearch(?SR) -> ChooseSearchRoute(?SR, ?ST)
Rule_3: WheeledRobots(?WR) ^ hasAbility (?A, ?WR) ^ Room(?ER) ^ hasPosition(?WR, ?ER) ^ SearchTask(?ST) ^ SquareSearch(?SR) -> ChooseSearchRoute(?SR, ?ST)

### 3.1.3. Robot Task Reasoning Based on JESS

In this section, JESS is adopted to map ontology knowledge into the internal format of inference engine and provide the tasks for robots to complete by matching the factual knowledge and rules. JESS is a java-based CLIPS reasoner and its core language is compatible with CLIPS [36]. The structure of JESS is shown in Figure 7. It includes three parts—Ontology, Rules Base, and Inference Engine. In Ontology, the facts library stores the initial fact information hierarchically. In the Rules Base, the rules library stores domain rules related to inference, and JESS internally uses the "if-else-then" grammatical structure to describe rules. Ontology provides the factual basis for inference, and the rules base provides inference rules. Since JESS inference engine cannot directly understand the knowledge of OWL and SWRL format, it first needs to analyze ontology and rules and convert them into the input format of the inference engine.

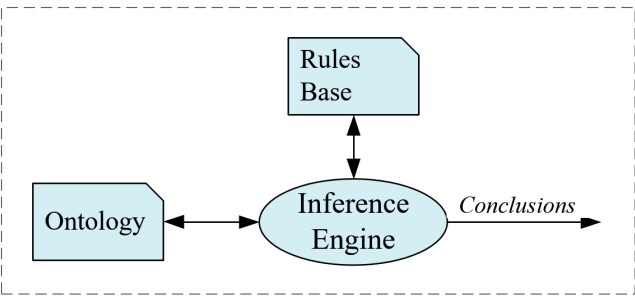

**Figure 7.** The structure of JESS.

When the real-time information in SAR is updated to the ontology, the inference engine can automatically complete the matching between ontology and rules. When the scene information meets the precondition of a rule in the rules base, the rule is called and the result of the rule is the task that the robot needs to complete in the current state.

### 3.2. Task Planning Algorithm Based on Ontology

Figure 8 interprets the representation method of atomic actions defined as the smallest granular actions which can be directly executed by a robot. Atomic actions are made up of execution preconditions and action effects. The former will have an effect on the robot and the state of the environment, such as deleting or adding some states.

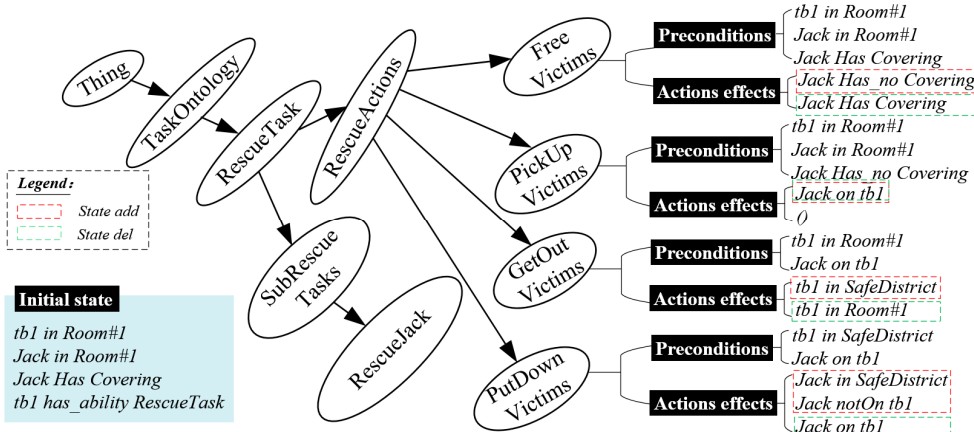

**Figure 8.** The representation method of atomic actions.

For a given high-level task, the program can adopt a specific search algorithm on the basis of the initial state of the robot and environment, which aims to match the preconditions and action effects of atomic actions and obtain the atomic action sequence of the specific task. When the robot performs a task for the first time, it will query and match the ontology layer by layer according to the current initial state and reason about the final atomic action sequence. The atomic action sequence contains the time sequence of action execution and is a complete representation of high-level input tasks. Then, the system will update the knowledge in the task ontology and write the complete representation of the high-level task into the definition of the task in the ontology. When the robot needs to perform the task next time, it can directly query the definition of the task to get the atomic action sequence corresponding to the task without querying and matching again. This improves the efficiency of task execution.

We proposed an ontology-based task planning algorithm whose example reflecting algorithm idea is shown in Figure 9. The robot knows that Jack is a victim and the current initial state. Besides, the robot can know the location, ability and state of robot tb1 by querying the ontology. Then, it is easy to see that the current initial state matches the preconditions of the search work. Thus, the robot first performs the subtask of *search Jack*. According to the subtask of *search Jack*, the robot matches the preconditions and states, and we can choose different search strategies to collect information about the victims. After obtaining Jack's information, it will update the current state by deleting the old state or adding a new one. Then, the robot performs the rescue task, as shown in Figure 8. The current state of the environment matches the precondition of the atomic action *Free Victim*, which means that the atomic action *Free Victim* can be executed first. After the *Free Victim* is executed, the environment state is updated. At this time, the environment state matches the atomic action *PickUp victim*, so the atomic action *PickUp victim* can be executed. By parity of reasoning, we can get a sequence of atomic actions: *CenterSearch tb1 Jack*, *CrossSearch tb1 Jack*, *Free Jack*, *PickUp Jack*, *GetOut Jack*, and *PutDown Jack*. The atomic actions are shown with the blue block in Figure 9.

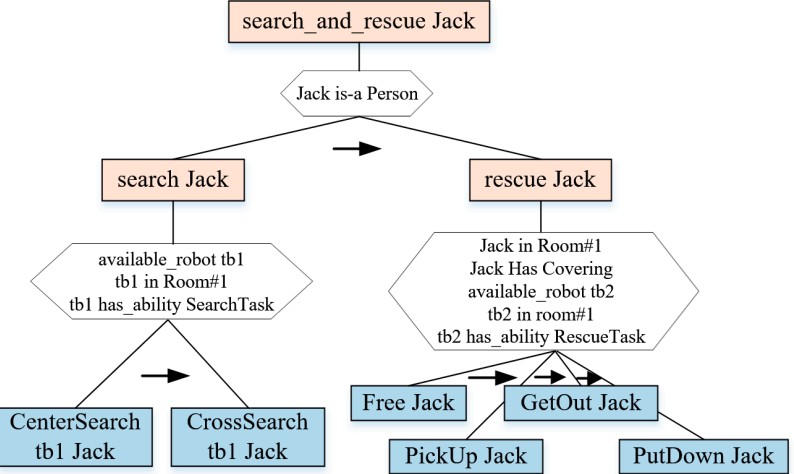

**Figure 9.** An example of task planning process.

The proposed algorithm 1 is shown as the pseudo-code below. Its input are the initial states, the task *t* reasoned by JESS, and the ontology knowledge *O*. The output of the algorithm is the sequence of atomic actions, which is also the plan for accomplishing the *t* from the initial state.

---

**Algorithm 1** An ontology-based task planning algorithm

---

**Input:** s: the initial state; *t*: the task reasoned by JESS; *O*: the ontology knowledge
**Output:** *P*: A plan for accomplishing the *t* from the initial state;

```
1:      procedure generate a plan for accomplishing the t
2:          P = the empty plan.
3:          function task_planning (t)
4:              if t is a primitive task then
5:                  modify s by deleting del(t) and adding add(t)
6:                  append t to P
7:              else
8:                  for all subtask in subtasks(t) do
9:                      if preconditions(subtask) matches the s then
10:                         task_planning (subtask)
11:         return P
12:     end procedure
```

---

### 3.3. The Update of Ontology

The objects in the disaster rescue scenario are represented as individual classes. Taking Figure 10 as an example, Jack is the instance of victim and has some attributes defined as we need. In the disaster rescue scenario, we pay more attention on the vital signs of victim such as heart rate (HR) and blood pressure (BP). In addition, the basic information of the victim is also vital for smart decision-making, such as position and gender. In the initial state, the position is default. The position knowledge will be updated with the corresponding victim being recognized and located, which is also the building process of the semantic map.

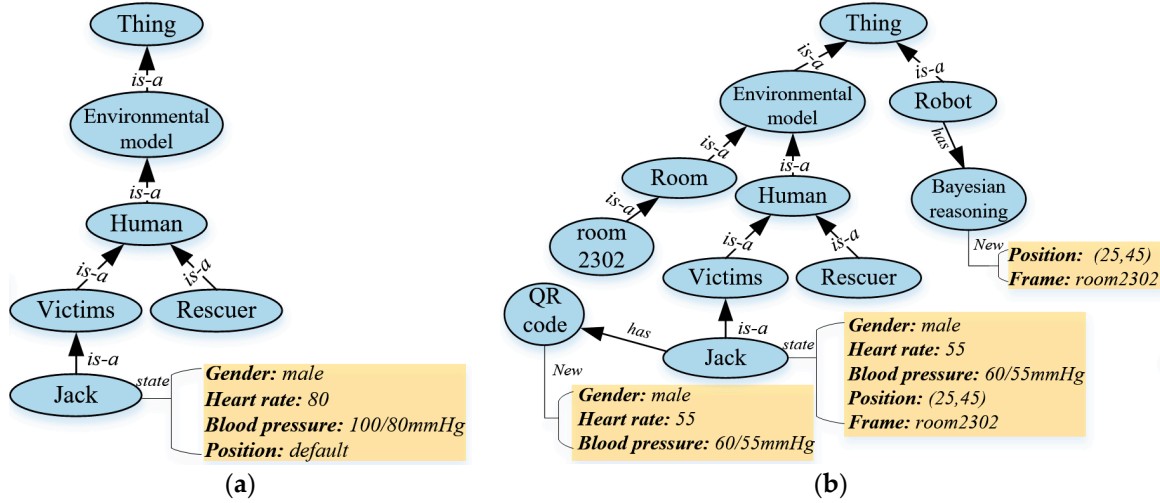

**Figure 10.** The knowledge representation in environment ontology. (**a**) The structure of environment ontology; (**b**) the knowledge update in environment ontology.

The traditional method to build the semantic map is to recognize and locate objects. Then the recognized information will update the environment ontology in the language of OWL. That connects the map by SLAM with the semantic information. In our research work, we make some assumptions and simplify to focus on the smart decision-making work. In the process of recognizing the objects in the disaster rescue environment, we adopt the method to recognize based on the Quick Response (QR) code, which avoids the complex semantic information obtaining process.

QR code is a 2D matrix code. It is widely used in daily life for its quick recognition ability, large and stable information reserve. In the process of the robot searching for the target, the QR code attached

to the simulated target can be scanned by the onboard camera. After the processing of the QR code recognition program, the functional semantic information can be obtained, providing the original data for the construction of the semantic map, shown in Figure 10b. According to the requirements of SAR tasks, the functional semantic information can be designed as the following form:

$$\theta = (GEN, HR, BP) \tag{3}$$

The target object is the victim ready to search to rescue. Thus, the QR code is designed to represent the victim. "GEN" is defined as the gender of victim, "HR" is defined as the heart rate of the victim, and "BP" represents blood pressure.

The robot recognizes the target through QR code and obtains semantic information. However, the position information of the target object, including room frame and concrete position information, is still missing. As shown in Figure 11, on the basis of previous research works, the position information of the target object is calculated by Bayesian reasoning.

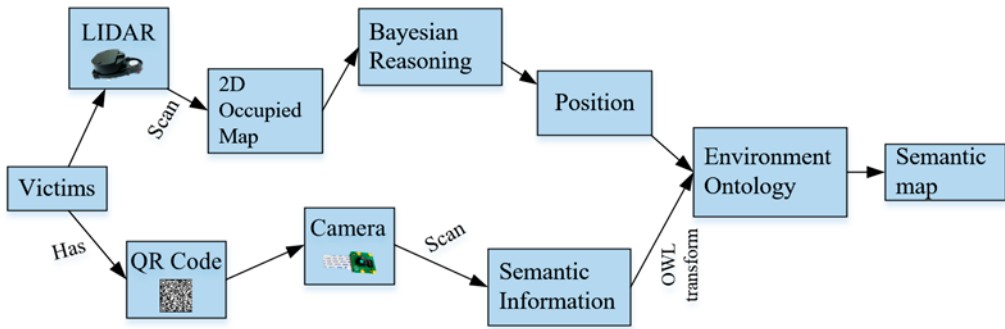

**Figure 11.** The process of semantic information recognition and updating.

$P_{G,t}(L)$ represents the probability of the existence of target object in grid $G$ at time $t$, $L$ represents the existence of target object, and then Bayesian reasoning can be used to continuously update the probability of detected target object. $P$ means that the target object does exist in the case that the algorithm detects the target, while $O$ means that the target object is detected. In addition, the existence of a target object at a certain position is also related to the geometric measurement information corresponding to this location, which can be obtained through the 2D occupied grid map constructed by the middle layer. As shown in Figure 12, $M(G)$ is used to represent the geometric categories of grid $G$, $\mathrm{M}(G) \in \{M1, M2, M3, M4\}$. $M1$ represents that the detected object is in the obstacle area of the grid map, $M2$ represents that the detected object is in the unknown area of the grid map, $M3$ means that the detection point is in the free area of the grid map but there are obstacles nearby, and $M4$ means that the detection point is in the free area of the grid map and no obstacles nearby. Besides, the definition of the size of the nearby area to set the threshold according to the specific environment, such as 20 cm.

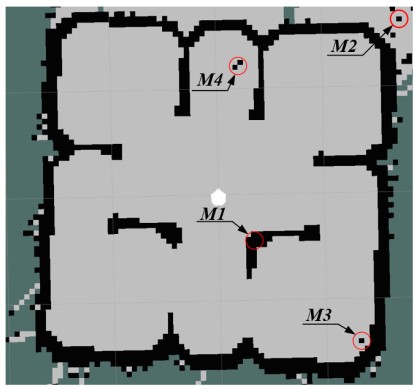

**Figure 12.** Definition of the four grid types.

The probability correlation of the geometric information of the grid and the target object in the grid can be expressed by prior knowledge $P(M(G)|L)$. After obtaining the semantic information of the QR code and the geometric categories of the grid, the posterior probability of the detected target object of the grid can be updated by Bayesian reasoning.

$$P_{G,t}(L|O_t, M(G)) = \frac{P(O_t, M(G)|L)P_{G,t-1}(L)}{P(O_t, M(G))} \tag{4}$$

Since the information of the two channels is independent, we can deduce

$$P_{G,t}(L|O_t, M(G)) = \frac{P(O_t|L)P(M(G)|L)P_{G,t-1}(L)}{P(O_t)P(M(G))} \tag{5}$$

In Equations (4) and (5), $P_{G,t-1}(L)$ is based on the priori knowledge of detecting object information on grid $G$ at all times before time $t-1$. We abbreviate the posterior probability $P_{G,t}(L|O_t, M(G))$ to $P_{G,t}(L)$ as the prior probability of the next iteration update. Therefore, at time $t$, event $O$ occurs in a grid $G$, that is, an object is detected, and the probability value of the grid is updated by using Equation (5). Similarly, for the inverse event $\overline{O}$, Equation (6) is used to update.

$$P_{G,t}(L|\overline{O}_t, M(G)) = \frac{P(\overline{O}_t|L)P(M(G)|L)P_{G,t-1}(L)}{P(\overline{O}_t)P(M(G))} \tag{6}$$

Finally, at the end of robot exploration, whether the final posterior probability of grid $G$ is greater than the set threshold can be used to determine whether the target object exists in the grid or not.

## 4. A Study Case Based on the Semantic Model

In this section, we take the high-level SAR tasks as the study case to study the high-level decision-making based on the ontology in Section 2 and the methods in Section 3.

### 4.1. Hardware and Software

As is shown in Figure 13, the hardware system in our system is established on TurtleBot3, which is a new generation of mobile robot platform established on ROS. Based on ROS, it is an ideal and essential platform to do research work.

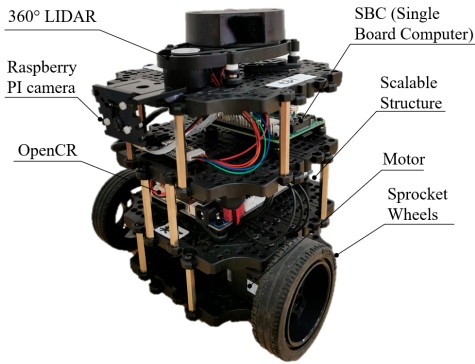

**Figure 13.** The TurtleBot3 Burger.

Accordingly, the software system is based on ROS, which is the most popular and vital middleware for robot system development. Figure 14 contributes our software system framework, which illustrates that the design and development of ROS nodes contributes the central part of the software system framework. Founded on ROS, the software system framework can be respectively classified into three control levels—control of actions, control of navigation, and control of velocity corresponding on the task planning module, navigation module, and base control module.

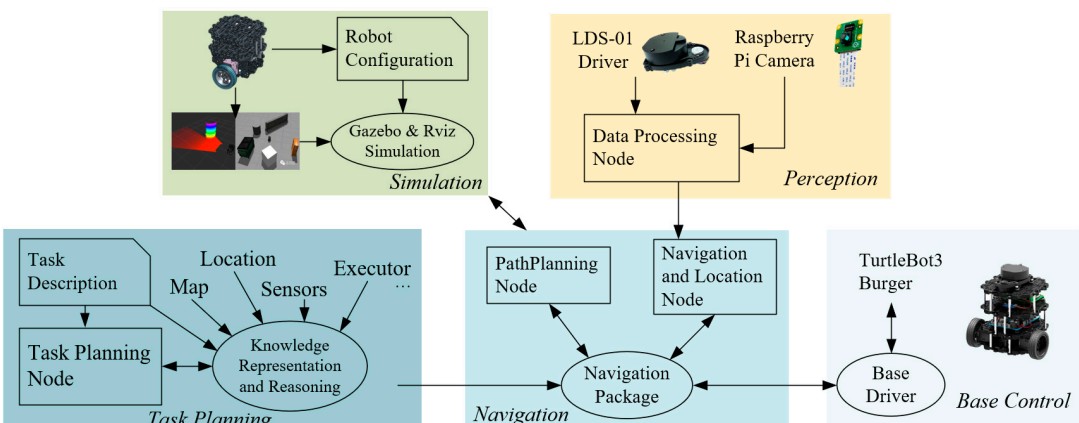

**Figure 14.** The framework of the software system.

*4.2. A Study Case*

An assumption is made in the study case that the TurtleBot3 is able to search, recognize, and rescue the victims. In addition, it is noteworthy that the adopted TurtleBot3 in the experiment only harbors mobility and obstacle perception. However, our research attaches importance to the decision-making at the related top level to the robot task planning and does not focus on such low-level control as the way to control movement and navigation. Therefore, we make a reasonable assumption that the robot is able to search, recognize, and rescue the victims.

Under the circumstances of the real robot experiment, the robot needs to search, recognize, and rescue the victims. The built experimental environment is shown in Figure 15a. The object elements in the environment are displayed in virtue of the label objects on the ground, for example, victims, books, and bookshelves. The environment map built through the LIDAR LDS-01 is shown in Figure 15b. The robot in our research work searches, recognizes, and rescues the victims based on the partially known environment. That is, the environment map and some semantic information are already known, but the exact state of the victim is unknown. In this case, the robot makes smart decisions based on the ontology and other algorithms.

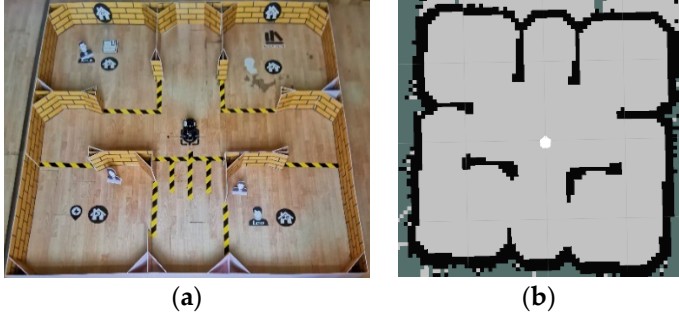

(**a**)                                (**b**)

**Figure 15.** The experimental environment we built: (**a**) The real experimental scenario; (**b**) The corresponding environment map built by TurtleBot3 in *Gmapping* algorithm.

The specific implementation process of the study case on TurtleBot3 is demonstrated in Figure 16. The sequence of atomic actions can be obtained by reasoning in the ontology knowledge base. The corresponding action properties, such as the target point and the time constraint of some actions, can be obtained in virtue of the analysis on the atomic action sequence and querying the ontology knowledge base. The robot then subscribes to the message and performs the corresponding actions through the navigation and path planning algorithm. Finally, the input high-level task is completed.

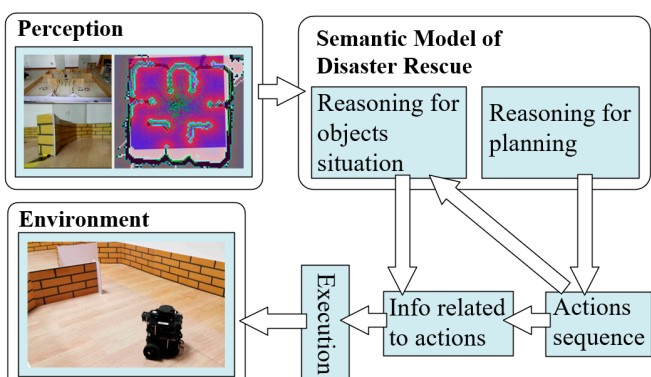

**Figure 16.** The specific implementation process of study case on TurtleBot3.

During the verification experiment, the communication among the devices is made through LAN, which shares a master computer and topic to realize the communication requirements between devices. As shown in Figure 17, ontology knowledge is stored in computer PC_2. Computer PC_1 undertakes the master computing. It is also the central computer employed to run the master node. Turtlebot3 tb_1 commits the program storage of map building, navigation, and path planning. It is worth noting that the map building algorithm adopts the *Gmapping*, the local path planning algorithm adopts the DWA, the global path planning algorithm adopts the D*, and the location algorithm adopts the *amcl*.

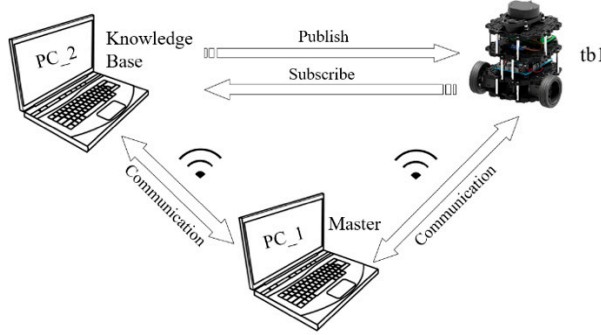

**Figure 17.** The communication among different devices.

## 5. Results

Figure 18 shows a sequence of snapshots for the real execution of the atomic actions sequence generated by the decomposition of the reasoned high-level task in a real world experimental scenario. A list of atomic actions obtained is displayed as navigating across the target points in a proper order according to action types. For example, the *Moveto* action needs to move to the specific target point.

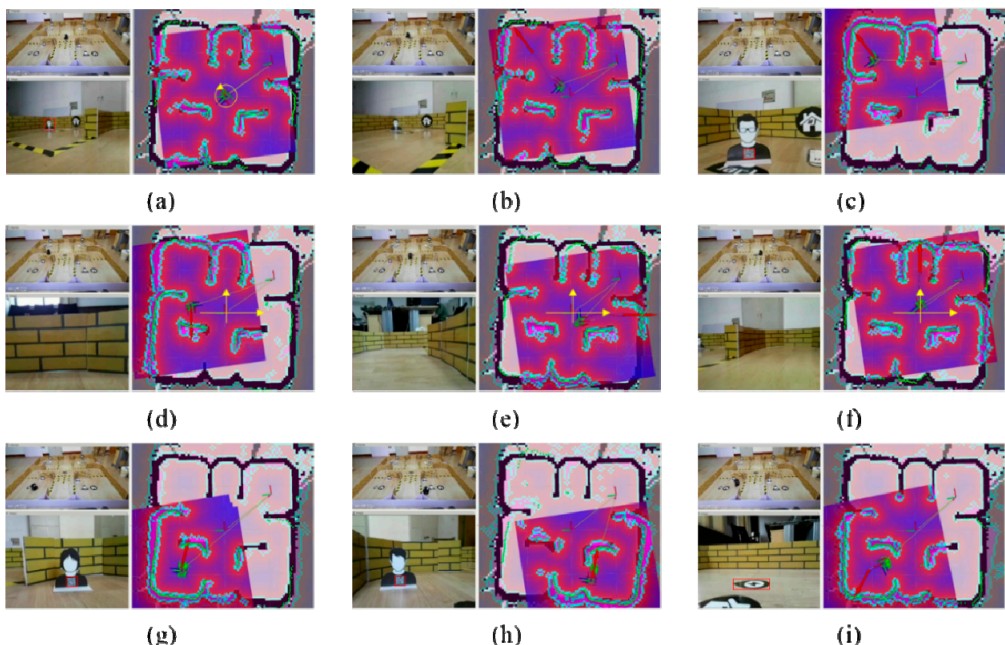

**Figure 18.** A sequence of snapshots for the real execution of the atomic action sequence decomposed from the rescue task. (**a**) The circle strategy to search the target object; (**b**–**d**) Find the target victim recognize and rescue the victim; (**e**,**f**) The cross-search strategy to find the other target victims; (**g**–**i**) Rescue the remain two victims and charge.

First, as shown in the Figure 18a, the TurtleBot3 is at the center of experimental scenario, thus the search strategy is turn around to search the target object. As shown in Figure 18b–d, the robot finds the target victim, moves to recognize the victim, and gets the victim out. Figure 18e,f shows the cross-search strategy. Figure 18g–i shows the robot searching for the other two victims and moving to the charge point to charge.

The experimental results show that the robot can make smart decisions based on the constructed ontology in SAR. The robot can perform different tasks with the different environment state. Then, according to the input task, the robot can decompose the task into a sequence of atomic actions. Besides, the robot can obtain the semantic information of victims by scanning the QR code. During the

experiment, the actions are simplified to retain the ability of movement and recognition. According to the atomic actions sequence, the robot passes through the target points successively to complete the task and achieve the final goal. In the process of task execution, it rescues the victims in turn according to the identified sequence. In the search strategy, different search strategies are adopted according to the initial position of the search.

## 6. Conclusions

In conclusion, an ontology in SAR scenario is built successfully for the smart decision-making of robots. The ontology in our article consists of entity ontology, environment ontology, and task ontology. Aiming at the core task ontology, we designed the representation method for high-level tasks and atomic actions in smart decision-making of robot, which facilitates the development of a subsequent task planning algorithm. In addition, the environment plays an important role in robot smart decision-making. Aiming at the environment ontology, the semantic model and the representation of objects in the environment are carried out.

The robot can perform suitable tasks to adapt to different environment states. In addition, the environment is known partially, and the environment ontology is constantly updated in the process of task execution to understand the environment. The task planning algorithm based on ontology is designed. Then, we have an experiment in the SAR scenario based on TurtleBot3. The experimental results show that the robot can successfully complete tasks and realize smart decision-making.

Future research work mainly focuses on the following aspects. First, the real SAR scenario is inaccurate, random, and incomplete. Therefore, it is necessary to study the task planning in uncertain environments. Then, multiple heterogeneous robots can cooperation to help with more complex tasks that a single robot cannot do. Therefore, it is of great significance to study cooperation task planning and its application based on multi-robot programming. Finally, the application of cloud-based knowledge will reduce the dependence of robots on specific hardware, which is conducive to the research of multi-agent and swarm intelligence.

**Author Contributions:** Conceptualization, Y.Z.; Funding acquisition, Y.Z.; Investigation, X.S.; Methodology, X.S.; Project administration, Y.Z.; Supervision, J.C.; Validation, X.S.; Writing—original draft, X.S.; Writing—review & editing, Y.Z. and J.C.

**Funding:** This research was funded by the National Natural Science Foundation of China (Grant No. 61806212, No. 61603403, No. U1734208 and No. 61702528).

**Conflicts of Interest:** The authors declare no conflict of interest.

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
