# Peer review of "High-Level Smart Decision Making of a Robot Based on Ontology in a Search and Rescue Scenario"

_futureinternet, doi:10.3390/fi11110230_

Round 1

Reviewer 1 Report

The paper proposes an ontology and an algorithm for task planning to get a sequence of atomic actions for search and rescue scenario. Authors evaluate the algorithm and ontologies using a real robot platform TurtleBot3.

The paper is generally good. It is not clear why authors propose the conceptual layer and instance layer as well as abstract layer and specific layer to represent the conceptual one. It should be discussed with the clarification.

Sofeware -> Software on the second ontology.

Reviewer 2 Report

The authors have made an attempt to provide a proof-of-concept for making smart decisions by the robot in the search and rescue situation. However, there are certain issues to be resolved before going to the next step.

1) Multi-agent systems features and methodologies is the core of crisis response systems. Crisis response systems take advantage of coordination and planning capabilities of multi-agent systems to handle response teams’ coordination and interaction problems.But, the multi-agent process is not defined properly to implement.

2)In the result section:  as mentioned "different search strategies are adopted according to the initial position of the search", It would be better to provide the results in a tabular form.

3) Where is the scenario for "move to recognize the victim and get the victim out".

4)For Eq.4 and Eq.5 provide the corresponding results.

5.What and where is the "Smart Decision" (Normally, "smart" is used for adaptability ".Hoe the system adapt to the new situations?

6)What about Spatial and Temporal reasoning aspects?

Reviewer 3 Report

The article is very well written and authors have made very well use of semantic technologies in search and rescue application. It would be excellent if authors can discuss and compare their experimental results with some benchmark dataset.

Reviewer 4 Report

This paper describes a set of ontologies created to help robots in their planning and decision plans.  I think the paper well written but it has many shortcomings that have to be corrected before it can be accepted for publication

In the paper, it is not clear why the authors need to use OWL ontologies for the desired task. The paper clearly says that since JESS inference engine cannot directly understand the knowledge of OWL and SWRL format, it first needs to analyse ontology and rules and convert them into the input format of the inference engine. Therefore, why the authors have decided to use OWL/ SWRL and not the input format required by JESS directly?. I think this must be clarified.

The ontologies and SWRL rules created should be published or at least attached to the paper for revision. I have many doubts about the representation of tasks such as the way the ordering is described, and if they are classes or they should be instances. Without the complete ontologies, it is not possible to verify their quality and suitability for the desired tasks.

Additionally some naming used for concepts show problems in the ontology that I think they must be corrected. For example, the concept names taskontology, robotontology and enviromentOntology are misleading as their subclases are not conceptually valid (eg. a robot cannot be a subclass of a robotOntology). There are also misspellings such as sofeware or the use of plurals.

The tasks descriptions in the models (ej: lines 154-160) should be described formally in OWL, this informal description is not suitable to understand what it is modelling.

In line 127 when the authors write about abstract layer and specific layer. I think is most common in the literature to talk about domain and application ontologies.

I think that what figure 6 describe are not communications are dependencies. They should be explained in more detail as it is not clear what all the act relations mean.

In section 3.2 it is described how that when the robot perform a task for the first time, it will query and match the ontology layer by layer according to the current initial state, and reason about the final atomic action sequence. This is remarked in the algorithm in page 10. However, this is confusing; inference models are commonly applied offline (they are precalculated). If you do this way, you would directly have the complete plan for the desired task (all the plans would have been extended with all the inferred information). In your context, why is it needed to do it on the fly the first time they are needed?

Round 2

Reviewer 4 Report

I think the paper can be accepted as it is.